# Engineered Oncolytic Adenoviruses: An Emerging Approach for Cancer Therapy

**DOI:** 10.3390/pathogens11101146

**Published:** 2022-10-04

**Authors:** Ee Wern Tan, Noraini Abd-Aziz, Chit Laa Poh, Kuan Onn Tan

**Affiliations:** 1Department of Biological Sciences, School of Medical and Life Sciences, Sunway University, No. 5 Jalan Universiti, Petaling Jaya 47500, Selangor Darul Ehsan, Malaysia; 2Centre for Virus and Vaccine Research (CVVR), School of Medical and Life Sciences, Sunway University, No. 5 Jalan Universiti, Petaling Jaya 47500, Selangor Darul Ehsan, Malaysia

**Keywords:** oncolytic adenovirus, immune system, tumor microenvironment (TME), cancer therapy

## Abstract

Cancer is among the major leading causes of mortality globally, and chemotherapy is currently one of the most effective cancer therapies. Unfortunately, chemotherapy is invariably accompanied by dose-dependent cytotoxic side effects. Recently, genetically engineered adenoviruses emerged as an alternative gene therapy approach targeting cancers. This review focuses on the characteristics of genetically modified adenovirus and oncology clinical studies using adenovirus-mediated gene therapy strategies. In addition, modulation of the tumor biology and the tumor microenvironment as well as the immunological responses associated with adenovirus-mediate cancer therapy are discussed.

## 1. Introduction

Cancer is one of the leading causes of human mortality worldwide. To date, chemotherapy is one of the most reliable treatments against cancers. Unfortunately, treatments with chemotherapeutic drugs are often associated with adverse side effects that are related to dose-dependent cytotoxicity. In addition, cancer metastasis and early recurrence remained significant among cancer patients after chemotherapy [1]. Therefore, alternative cancer therapeutics, including virus-mediated gene therapy, are well sought after to increase clinical benefits and minimize treatment-associated off-target effects. 

Oncolytic viruses (OVs) are viruses that target and destroy cancer cells while minimizing the negative impact on normal cells. The anti-tumor efficacy of OVs is achieved by direct viral lysis of cancer cells and stimulation of the host’s anti-tumor immunity. The dissemination of infectious OVs from lysed tumor cells to adjacent uninfected tumor cells amplifies the oncolytic activity of the OVs [2]. Following the approval of Imlygic^®^ (Talimogene laherparepvec; T-VEC) by the US Food and Drug Administration (FDA), OVs are gaining substantial attention as the first class of anti-cancer treatments. This is evidenced by the fact that over 40 OVs are now being evaluated in clinical studies as monotherapy or combination treatment for a variety of malignancies [3]. 

Virus-mediated gene therapy consists of the construction of a gene expression construct with the therapeutic gene in the viral vector, followed by the generation of the recombinant viral particles in human packaging cell lines. The recombinant viral particles are highly effective in mediating gene expression in highly permissive human cells, inducing the expression of therapeutic genes in the host’s cancerous cells to enhance oncolytic effects [4]. In addition, viral-mediated gene therapy can be developed to induce cytotoxic T cells’ response to enhance immunity against tumor cells [5]. Oncolytic adenovirus (OAV) is one of the most extensively studied OVs due to its high potential in oncolysis and immune response stimulation. Immunostimulatory transgenes may be readily inserted onto OAVs to modify the tumor microenvironment (TME) and stimulate the immune effector responses.

## 2. Adenovirus: Structure, Genome, and Serotype

Adenovirus is formed of an icosahedral protein shell measuring 90 nm and harboring linear double-stranded DNA (dsDNA) that belongs to the genus Mastadenovirus of the Adenoviridae family [6]. The icosahedral capsid outer shell is composed of three primary types of proteins including hexon, penton, and fiber [7]. Interestingly, the hexon, which comprises the majority of the adenovirus capsid, is made up of up to 240 homotrimers to encapsidate the adenovirus [7]. In addition, the fibers arise from the 12 vertices of the icosahedron, whereas the penton base is located at the base of each fiber [8]. The genome of the virus encodes approximately 35 proteins that are produced in two phases, the early and late phases. The early phase corresponds to the start of viral DNA replication seven hours after infection, whereas the late phase refers to the incidence of post-DNA replication. The first 20 synthesized proteins have regulatory functions that allow the virus to gain control of the cells, followed by viral DNA replication activity.

In contrast, the proteins synthesized at a later stage are mainly structural proteins of the virus [9]. As early as one day post-infection, the virions are assembled at the nucleus of the host cell, and for several days, the host cells lyse and release the infectious virus. As reported in a previous study, more than 57 serotypes and over 100 genotypes of human adenoviruses consist of 7 species from A to G (HADV-A to HADV-G) [9]. Species A, B, C, D, E, and F are widely circulating and have been associated with human infection outbreaks [6,10]. Serotype refers to the ability of specific antisera to neutralize infected cells. Despite the diversity of its serotype, the structure and protein activities of all serotypes are similar. However, specific protein activities lead up to the distinctive characteristics of each serotype. For instance, the packaging domain of the adenovirus is serotype specific [11]. In a previous study, it was reported that IVa2 and 22K proteins complement the activity of their corresponding human adenovirus C5 protein, but not L1 52/55K protein of human adenovirus 17 serotypes [12]. This suggests that 52/55K protein is important for serotype specificity of adenovirus DNA packaging. 

## 3. Adenovirus: Pathogenicity and Immunogenicity

Most people have been infected by adenovirus, leading to lifelong immunity. In general, adenovirus species C, consisting of Ad 1, 2, 5, and 6, is the most common to infect humans, especially in early childhood, consequently causing infection of the upper respiratory tract [13]. In addition, adenovirus species B, such as Ad 3, 7, 11, 14, 16, 21, 34, 35, and 50, contributes to 5% of common cold cases, involving infection of the upper respiratory, gastrointestinal, and urinary tract [9]. In most adenovirus infections, the symptoms are minor, but they might be fatal for immunocompromised individuals [9]. However, despite low pathogenicity, some adenoviruses result in severe symptoms, especially in children. For instance, human adenovirus (HAdV) types F40 and F41 are a leading cause of diarrhea and diarrhea-related mortality in infants and toddlers worldwide [14]. Adenovirus can be propagated, replicated, and grown to achieve high viral titers of up to 1 × 10^13^ virus particles per mL [15]. Immunocompetent individuals can develop minor, self-limiting clinical pathologies due to the immunogenicity of adenovirus. The host immune response to adenoviral infection limits the use of adenovirus in gene therapy; however, immune stimulation such as the induction of a systemic pro-inflammatory state, the recruitment of cytotoxic immune cell populations to the infection sites to eradicate virus-containing cells, and the alerting of adjacent uninfected cells of viral infection is beneficial for the development of immunotherapies against cancer [16]. Additionally, oncolytic adenoviruses are designed to precisely deliver therapeutic transgenes to tumor cells without affecting healthy cells [17]. Both replicative and nonreplicating adenoviruses (Figure 1) are used as in situ cancer vaccines due to their high immunogenicity [18]. 

### 3.1. Innate Immune Responses

Due to pathogen-associated molecular patterns (PAMPs) components of the viral capsid and viral nucleic acids, adenovirus infections elicit an innate immune response comprised of cellular components such as pattern recognition receptors (PRR) [18]. The interaction of adenovirus fiber and knob capsid proteins with the coxsackie and adenovirus receptor as well as the v integrins induces activation of nuclear factor-kB, leading to the expression of chemokines and interleukins (ILs) [19]. In addition, according to a prior study, an innate immune response to capsid proteins is elicited immediately after intravenous administration in a murine model for up to six hours [9]. However, the response was lethal at high dosages of more than 10^10^ viral particles. This leads to the stimulation of biphasic synthesis of pro-inflammatory proteins, including IL-6, TNF-α, IFN-γ, IL-1β, and IL-12, along with chemokines [9]. 

In addition, following viral entry, PRRs include Toll-like receptor (TLR) 9 in endosomes, and cytosolic sensors such as DNA-dependent activator of IFN-regulatory factors (DAIs) or cytosolic inflammasome (NALP3) recognize viral DNA [20,21,22]. For this, the interferon response is stimulated [23]. The interferon response eliminates the virions from the cells and inhibits the E1A transcription, subsequently blocking the replication of adenovirus [20,24].

### 3.2. Adaptive Immune Responses

Adenovirus infection that causes upper respiratory infections induces anti-adenovirus serotype-specific antibodies and cross-reactive T cell responses [25]. As reported in a previous study, serotype 5 is known to be the most prevalent adenovirus strain, which has a seroprevalence of approximately 50% in North America and close to 100% in Africa [26]. In addition, the seroprevalence of adenovirus serotype 35 ranges from 3% to 22% in the United States [26]. 

Adenovirus is strongly immunogenic, consisting of the three major capsid protein antigens, which are the hexon, penton base, and fiber. The majority of neutralizing antibodies are targeted against the capsid proteins and non-conserved loops that differ across serotypes and are exposed on the surface of hexon [27]. Interestingly, the presence of adenovirus-specific CD4+ T cells in peripheral blood lymphocytes has been reported in nearly all individuals of all ages, where the CD4+ T cell epitopes are in the hexon region [28]. In addition, cytotoxic T lymphocytes (CTL) specific to the hexon were also reported to be protective against adenovirus infections in humans [29].

## 4. Genetically-Modified Adenoviruses for Cancer Therapy

The majority of genetically engineered adenovirus vectors are serotype 5, either replication-defective or replication-competent adenovirus (Figure 1), and have been studied extensively (Table 1). Replication-competent adenoviruses such as OAV can replicate and lyse tumor cells without harming healthy cells, as OAVs were mainly developed to facilitate efficient tumor cell lysis and virus spreading, hence providing reliable control of viral replication in normal cells [3]. For this, the therapeutic effect is amplified by virus replication and reinfection of adjacent tumor cells [30]. Replicative adenovirus is reported to be more effective for gene therapy [31]. Kim et al. have reported that immunocompetent mice with intracranial glioma require replication-competent adenoviral vectors to elicit a significant antitumor immune response and lifespan advantages compared to replication-defective adenovirus [32]. 

Replication-defective vectors are genetically modified by deleting E1A and E1B genes (Figure 1), resulting in defective virus replication and inhibition of the host cell’s apoptotic response to adenovirus infection. E1A encodes the E1A protein required for adenovirus replication, which induces the expression of E1B, E2, E3, and E4 transcription units that affect the expression of a large variety of cellular genes to promote adenovirus replication [9]. Adenovirus infection with a deletion in the E1 gene does not cause a cytotoxic effect; therefore, it is approved to be administered in humans for gene therapy [33]. Most of the recombinant adenoviruses used now are E1 deleted [9]. These deleted genes are replaced by an expression cassette harboring a highly active promoter, such as the cytomegalovirus immediate-early (CMV) promoter, that induces the expression of the foreign transgenes. In addition, vectors are constructed from plasmid or adenovirus DNA and are maintained in cell lines such as HEK293, PER.C6, or N52.E6 to express the E1A and E1B genes [34]. The incorporation of the transgene in the adenovirus genome allowed the adenovirus-mediated gene expression to be investigated in human cells, including cancer cells, as targeted cancer therapy (Table 1). However, recent development in adenovirus technology have led to the generation of a gutless adenovirus that can accommodate up to 36 kb transgenes without the use of helper adenovirus in its production, and immune response against the virus is likely to be minimized [35]. The large transgene capacity of gutless adenovirus allows the expression of multiple transgenes in an expression construct such as the bicistronic or tricistronic expression of transgenes [36,37] In addition, the high-capacity adenovirus is categorized as a third generation of replication-defective adenovirus, which is known as helper-dependent or gutless adenovirus [38]. High-capacity adenovirus possesses certain characteristics such as genetic stability, excellent in vivo transduction efficiency, and high titer production. Unlike the first and second generations of replication-defective adenovirus, high-capacity adenovirus only retains short non-coding regions from the adenovirus genome, such as ITRs and ψ signal [38]. With this respect, the absence of viral coding regions in the genomes of high-capacity adenovirus increases the cloning capacity to 36 Kb and permits the long-term episomal survival of transgenes in non-dividing cells.

**Table 1 pathogens-11-01146-t001:** Adenovirus-mediated gene expression targets various types of cancer.

Targeted Therapy	Type of Adenovirus	Gene Delivery	Route of Administration	Reference
**Carcinoembryonic antigen (CEA)**	Replication-defective	ETBX-011 containing epitope CAP1-6D (Ad5-CEA)	Subcutaneous	[39]
**Prostate cancer**	Replication-defective	Human prostate-specific antigen (PSA)	Subcutaneous	[40]
**Human papillomavirus (HPV)**	Replication-defective	Combination of Ad5-E6E7 and oncolytic maraba virus MG1-E6E7	Intravenous	[41]
**Melanoma**	Replication-defective	Melanoma-associated antigen A3 (MAGEA3)	Subcutaneous	[42]
**Brain tumour and glioblastoma**	Replication-defective	Transgenic human IL-12 p70 under the control of the RheoSwitch Therapeutic System (RTS) (Ad-RTS-hIL-12)	Oral	[43]
**Ovarian Cancer; Colorectal Cancer**	Replication-competent	GM-CSF	Intravenous	[44]
**Colorectal, ovarian, pancreatic cancers**	Replication-competent	TMZ-CD40L and 41BBL biliary	Intratumorally	[45]
**Ovarian cancer; peritoneal** **cancer**	Replication-defective	p53	Intraperitoneal	[46]
**Melanoma**	Replication-competent	TNF-alpha and IL-2	Intravenous	[47]
**Prostate Cancer**	Replication-competent	ORCA-010	Intratumorally	[48]
**Glioma**	Replication-competent	NSC-CRAd-S-pk7	Intratumorally	[49]
**TNBC *; Lung Cancer**	Replication-competent	ADV/HSV-tk	Intratumorally	[50]
**Breast, prostate, and testicular cancers**	Replication-defective	REIC/Dickkopf-3 (Dkk-3)	Intratumorally	[51]
**Pancreatic Cancer**	Replication-defective	REIC/Dickkopf-3 (Dkk-3)	Intra and peritumorally	[52]
**Hepatocellular carcinoma**	Replication-defective	SGE-REIC	Subcutaneous	[53]
**Biliary cancer**	Replication-defective	SGE-REIC	-	[54]
**Bile duct Cancer**	Replication-defective	SOX17	-	[55]

* TNBC—Triple-Negative Breast Cancer.

### 4.1. Tumor Carcinoembryonic Antigen (CEA)

An adenovirus vaccine serotype 5 encoding modified carcinoembryonic antigen (CEA), known as ETBX-011, which contains the epitope CAP1-6D (Ad5-CEA), carcinoembryonic antigen Peptide 1–6D, has progressed in several phase I and II trials [39]. This modified Ad5-CEA lacked the E1-, E2b region, and its expression is regulated by the CMV promoter to induce CEA-specific cell-mediated immune responses with anti-tumor activity. This Ad5-CEA virotherapy was reported for patients with metastatic colorectal cancer, and CEA-directed T cell responses were elicited, resulting in an 11-month overall survival advantage [56]. However, the real impact of this novel approach on survival rates will not be determined until a randomized, controlled trial with a greater number of patients is conducted.

### 4.2. Prostate Cancer

The efficacy of the replication-defective adenovirus serotype 5 encoding human prostate-specific antigen (Ad5-PSA) was evaluated in prostate cancer [40]. This recombinant adenovirus was reported to induce an anti-PSA T cell response, leading to the destruction of PSA-secreting tumor cells in a pre-clinical mouse model. This approach has advanced to phase I and phase II clinical trials, in both of which patients exhibited anti-PSA T cell responses [57,58]. 

### 4.3. Human Papillomavirus (HPV)

MG1-E6-E7, consisting of oncolytic maraba virus strain MG1 vaccine and replication-defective adenovirus encoding human papillomavirus (HPV) genes E6 and E7 (Ad5-E6E7), was evaluated in HPV-positive cancers followed by sequential treatment with atezolizumab, an anti-PD-L1 antibody [42]. The results showed tumor-specific responses, which significantly prolonged survival in the pre-clinical study and progressed to phase 1 clinical trial with atezolizumab in HPV-associated cancer patients [41,42].

### 4.4. Melanoma

Patients with advanced MAGE-A3-positive solid tumors were administered melanoma-associated antigen A3 (MAGEA3) using adenovirus in the clinical trial [42]. Furthermore, a previous clinical trial was also reported to assess oncolytic MG1-MAGEA3 with Ad-MAGEA3 vaccination in combination with pembrolizumab, an anti-PD-L1 agent, in patients with metastatic non-small cell lung tumors that were previously treated [42,57].

### 4.5. Adenoviruses Expressing Immune Modulators and Cytokines

Adenovirus was manipulated to produce immune modulators and cytokines. For instance, OVs were utilized to express granulocyte-macrophage colony-stimulating factor (GM-CSF) in promoting dendritic cell migration and maturation, consequently activating T cells (Table 2) [59]. In addition, adenovirus armed with IL-12 demonstrated an anti-tumor effect and enhanced immune stimulation in pre-clinical and clinical trials (Table 2) [59,60]. Another study demonstrated that adenovirus-enhanced IL-12 reduced the toxicity of systemic accumulation by cleaving the N-terminus of IL-12 peptides that were not released but were delivered directly to the tumor [61]. 

Ad-RTS-hIL-12 is a replication-defective adenovirus harboring the human IL-12 p70 transgene under the control of the RheoSwitch Therapeutic System (RTS). Veledimex, an activator ligand, is essential for the transcription of the IL-12 transgene. This approach has advanced to a phase I clinical trial involving patients with brain tumors and glioblastoma [43]. Other interleukins, including IL-2, IL-24, and IL-13, have been utilized to arm oncolytic adenovirus and have shown immune-stimulating capabilities in cancers [62].

**Table 2 pathogens-11-01146-t002:** Armed oncolytic adenovirus expressing immune modulators and cytokines in clinical trials.

Oncolytic Adenovirus	Type of Adenovirus	Backbone	Transgenes	Type of treatment	Reference/Identifier	Phase status	Reference
**CG0070**	Replication competent	Ad5	GM-CSF	Safety and effectiveness of CG0070 in high-grade non-muscle invasive bladder cancer patients who failed BCG treatment.	NCT02143804	Phase II (Withdrawn: Change in trial design)	[63]
**ONCOS-102**		Ad5; Ad3 fiber knob	GM-CSF	Open-label trial of GM-CSF coding oncolytic adenovirus CGTG-102 with low-dose cyclophosphamide in patients with refractory solid tumors.	NCT01598129	Phase I (Completed)	[64]
				To assess the safety of DCVAC/PCa with ONCOS-102 in males with castration-resistant advanced metastatic prostate cancer who have progressed after initial hormone or chemotherapy treatment.	NCT03514836	Phase II Terminated (insufficient accrual)	NA
	Replication competent			Dose escalation and dose expansion study of GM-CSF encoding adenovirus, ONCOS-102, in combination with anti-programmed death ligand-1 (PDL1) antibody, durvalumab, in adults with peritoneal disease who have failed prior standard chemotherapy and have platinum-resistant or refractory epithelial ovarian cancer or colorectal cancer.	NCT02963831	Phase I/II (Completed)	[44]
				Determine the safety, tolerability, and effectiveness of ONCOS-102 with chemotherapy.	NCT02879669	Phase I/II (Active, not recruiting)	[65]
				Sequential ONCOS-102 and pembrolizumab therapy safety.	NCT03003676	Phase I (Completed)	[66]
**Ad-RTS-hIL-12**	Replication defective	Ad5	IL-12	Safety and tolerability of a single Ad-RTS-hIL-12 tumor injection with oral veledimex.	NCT02026271	Phase I (Completed)	[43]
**LOAd703**		Ad5	4-1BB CD40L	To evaluate LOAd703 in pancreatic, biliary, colorectal, or ovarian cancer patients with conventional chemotherapy or gemcitabine immune-conditioning.	NCT03225989	Phase I/II (Recruiting)	[67]
	Replication competent			To determine whether intratumoral LOAd703 injections can reduce tumor growth and increase patient survival.	NCT02705196	Phase I/II (Recruiting)	[68]
				To assess the safety and efficacy of delolimogene mupadenorepvec (LOAd703) and atezolizumab in melanoma patients.	NCT04123470	Phase I/II (Recruiting)	NA
**TILT-123**	Replication competent	Ad5; Ad3 fiber knob	TNFα-IRES-IL-2	To study the safety of oncolytic adenovirus TILT-123 as monotherapy and with TILs in metastatic melanoma patients.	NCT04217473	Phase I (Recruiting)	[47]
**DNX-2440**	Replication competent	Ad5-delta24	OX40L	First or second GBM recurrence patients will be treated with DNX-2440.	NCT03714334	Phase I (Recruiting)	NA
**NG-641**	NA	Ad11/3	IFNα, CXCL9, CXCL10, FAP-BiTE	Safety and tolerance of NG-641 in metastatic or advanced epithelial tumor patients.	NCT04053283	Phase I (Recruiting)	[69]
**NG-350A**	NA	Ad11/3	CD40 agonist mAb	Evaluating the safety, tolerability, preliminary effectiveness, pharmacokinetics, immunogenicity, and other pharmacodynamic effects of NG-350A in patients with advanced or metastatic epithelial tumors.	NCT03852511	Phase I (Completed)	[70]

Abbreviations: BCG, Bacillus Calmette-Guerin; FAP-BiTE, Fibroblast Activation Protein-Bispecific T cell Engager; GBM, glioblastoma; GM-CSF, Granulocyte Macrophage Colony Stimulating Factor; IFNα, interferon alpha; IL-12, interleukin 12; mAb, monoclonal antibody; PCa, prostate cancer; TIL, tumor-infiltrating lymphocytes; TNFα, tumor necrosis factor alpha; NA, Information not available.

## 5. Strategies for Modulation of the Tumor Microenvironment (TME) 

### 5.1. Characteristics of TME

TME is critical for the development or suppression of tumor progression. TME is composed of cellular components such as tumor cells, tumor stromal cells, immune cells, and the extracellular matrix (ECM) of non-cellular components [71]. These components of TME together constitute an environment that allows tumor cells to continue to thrive. As the tumor develops, tumor cell metabolism is affected, leading to hypoxia, oxidative stress, and a low pH value TME, which is loaded with immune cells and soluble cytokines. The resulting localized hypoxia and low pH microenvironment may increase angiogenesis, stimulate tumor growth factors, potentially inhibit tumor cell apoptosis, and increase the resistance of tumor cells to standard therapeutic modalities, including radiation, cytotoxic drugs, as well as immunotherapy [72]. Immunological phenotypes of the TME are mainly categorized into three primary categories: (1) immune desert TMEs, (2) immune exclusion TMEs, and (3) inflamed TMEs [73]. Inflamed TMEs, also known as “hot” tumors, are characterized by an abundance of cytokine-secreting CD4+ and cytotoxic CD8+ T cells, myeloid cells, and monocytes in the tumor core. Inflammatory tumors are frequently associated with cancer immunotherapy patient responses [74]. The excluded TMEs are also very prevalent in immune cells, even though the immune effector cells are scarce in the tumor core. Due to the absence of particular chemokines and the availability of major barriers or specific inhibitors, it is hypothesized that these immune cells are mostly localized in the tumor’s periphery [75]. The deserted TMEs are classified as immunologically "cold" tumors, which are often characterized by the absence of immune cells and cytokines in the tumor’s core or stroma. Considering TMEs as the “regulator” and “target” when developing and selecting therapeutic genes shows significant benefits for enhancing the accuracy and efficiency of cancer gene therapy [76].

### 5.2. Modulation of TME with OAVs and Small Molecule Drugs

Solid tumor development requires tumor cell and microenvironment collaboration. The intimate interaction between viruses and the immune system can strengthen the immunological compartment of the TME. Due to a high level of antiviral defense conservation within the adaptive immune system, virus-infected tumor cells will become more susceptible to destruction by neighboring immune cells. Thereafter, virus-mediated oncolysis occurs, resulting in the generation of pathogen-associated molecular patterns (PAMPs) and danger-associated molecular patterns (DAMPs), which are both well-established sensors of immunogenic cell death [77]. Moreover, the apoptosis of tumor cells enhances the availability of tumor antigens, facilitating CD4+ and CD8+ T lymphocytes to be activated by dendritic cells [78]. 

In addition to the intrinsic ability of viruses to substantially affect the TME, it is also possible to induce tumor cells to produce therapeutic proteins before death [79]. This is accomplished by including transgenes that encode pro-apoptotic or tumor suppressor proteins in the oncolytic virus’s genome [80].

Interference of anti-tumor immunity by adoptive T cell therapy was a pioneering approach to achieving significant and long-lasting therapeutic advantages in patients with solid metastatic malignancies by employing patient-derived tumor-infiltrating lymphocytes (TILs) that were triggered and grown in large numbers before being reinfused into the patient [81]. Generally, T cell infusion is accompanied by lymphodepleting preconditioning with high-dose chemotherapy to target suppressive cells and increase the availability of systemic and intratumoral cytokines essential for T cell transplant activity [81].

In addition, TIL-based adoptive cell transfer has exhibited significant therapeutic outcomes with high objective response rates. Interestingly, the efficacy of TIL immunotherapy is associated with or enhanced by viruses [82]. Therefore, targeting primarily intratumoral T cell rejuvenation to promote cancer cell death is a sensible method for enhancing the immunogenicity of solid tumors. For example, IL-2 and tumor necrosis factor-alpha (TNF-α) are two cytokines with these characteristics.

The possibility of T cells being entrapped in stromal barriers or inactivated when exposed to an immunosuppressive microenvironment is a key issue that could compromise the efficacy of conventional TIL transfer. Additionally, OAVs expressing TNF-α and IL-2 and other cytokines or chemokines (highlighted in Section 5.3 and Section 6.1) can reduce local immunosuppression by increasing the production of pro-inflammatory components essential for T cell maintenance and activating costimulatory markers on dendritic cells in substitution of lymphodepleting preconditioning. Therefore, by substituting lymphodepletion with the virus, effectiveness was not only preserved but enhanced [62]. 

To extend the therapeutic advantages of adoptive cell therapy to a wider proportion of patients, pre-clinical research has shown that the immunological characteristics and engineering adaptability of OVs can be exploited to reprogram the TME. However, before the clinical translation of these strategies, it is crucial to analyze these data in considering the microenvironmental heterogeneity across tumor types, the origin and location of the tumor, and whether it is a subcutaneous or metastatic tumor which can have an impact on the tumor’s microenvironment [83].

To facilitate the penetration of the adenovirus into the tumor, the protective tumor microenvironment can be weakened or destroyed by small molecule inhibitors that target specific features of the TME, including the hypoxia and acidic environment of the TME that supported the growth and progression of the tumor cells [84,85,86,87]. Moreover, the TME is enriched in cancer-associated fibroblasts (CAF), which secrete growth factors, cytokines, and other protein factors that promote the growth of the tumor cells. In addition, endothelial cells (EC), which play an important role in the angiogenesis of blood vessels in the tumor, support the tumor cells growth. Thus, adenovirus-mediated gene therapy is likely to be efficacious without these tumor-promoting cells. In addition, it has been reported that drugs targeting CAF and EC exhibit potency in the growth inhibition or migration of these cells [88,89,90,91,92]. The TME is fortified with extracellular matrix proteins consisting of collagen, fibronectin, and other fibrous proteins which might hinder the penetration of the adenovirus into the tumor microenvironment. Specific drugs to inhibit the activity of enzymes involved in the synthesis of these proteins or remodeling of extracellular matrix were reported [88,93,94,95].

### 5.3. Strategies for Using Armed OAVs to Target the TME

Effective oncolytic virotherapy is hampered by immunosuppressive TME. Generally, tumor cells release immunosuppressive chemokines and cytokines, hence minimizing the amplitude of anticancer immune responses induced by OVs. Over the past two decades, gene therapy has made enormous progress. Early cancer gene therapy research mainly concentrated on delivering tumor suppressor genes, primarily wild-type p53, but subsequent efforts used the tumor phenotype to engineer a targeted cytotoxic effect. Thus, gene therapy strategies that modulate the relatively genetically stable TME, including stimulation of the local immune system, have recently been established. For instance, progress has been produced in depleting cancer stroma, degrading tumor vasculature, and boosting the anti-tumor response or suppressing tumor tolerance with cytokine-based gene therapies [96,97,98]. In addition, armed OVs are becoming highly desirable as therapeutic agents as they drive tumor-specific oncolytic cell death by lysing tumor cells to disclose PAMPs and DAMPs as well as tumor antigens and expressing encoded immunomodulatory transgenes. The TME could be modulated by OVs to make it less immunosuppressive. 

To resolve the immunosuppressive TME, various OVs, including adenovirus, have been genetically engineered to express GM-CSF to promote the development and differentiation of innate immune cells, hence triggering immunostimulatory signals within the TME [78]. In addition, several studies have engineered OVs to express immune-activating cytokines (such as interleukin) or chemokines, promoting T cell proliferation, differentiation, and effector activity in the TME [99]. Moreover, some OVs have been designed to express T cell costimulatory molecules, including CD40, 4-1BB, or OX40 (CD134), to stimulate tumor-specific T cell activation and strengthen anti-tumor immunity [100,101,102,103,104].

## 6. Strategic Design of OAVs Targeting Tumors

### 6.1. OAVs Armed with Cytokines and Chemokines

Cytokines are soluble proteins that facilitate cell-to-cell communication and maintain immune system homeostasis. They play a crucial function in the regulation of the innate and adaptive immune systems. Cytokines can inhibit tumor cell proliferation in the TME by anti-proliferative and pro-apoptotic activity or detection by cytotoxic effector cells. Several cytokines, including GM-CSF, IL-2, IL-12, IL-15, IFN-α, IFN-β, and IFN-γ, have shown anti-tumor properties in pre-clinical studies as well as in clinical trials. Among these cytokines, GM-CSF, IL-2, IL-12, TNF-α, and IFN-γ have been the most extensively investigated. 

One of the most prevalent cytokines used to arm OVs is GM-CSF. Different cell types, including activated T and B cells, macrophages, fibroblasts, and endothelial cells, have been shown to secrete GM-CSF [105]. GM-CSF aids antigen presentation through the recruitment and activation of DCs and macrophages [106]. The incorporation of a GM-CSF transgene into the adenoviral genome elevated DC activity, hence enhancing tumor antigen presentation to T cells [78]. For instance, CG0070 and ONCOS-102 are two OAV5 armed with GM-CSF in clinical trials [107]. In a phase I trial of CG0070, 35 patients with non-invasive bladder cancer who had previously failed bacillus Calmette-Guerin (BCG) treatment were enrolled. The median duration of treatment for patients was 10.4 months, with a response rate of 48.6%. The overall response rate among patients who received numerous intravenous infusions of CG0070 increased to 63.6% [63]. ONCOS-102 (Ad5/3-D24-GMCSF), a novel OAV encoding GM-CSF, effectively eradicated human melanoma cells and achieved total tumor regression in a nude mice xenograft model [107]. Virus-expressed GM-CSF promoted the differentiation of monocytes into macrophages. Due to encouraging results in pre-clinical experiments, ONCOS-102 was evaluated in a phase I clinical trial that involved 12 patients with refractory solid tumors (NCT01598129) [64]. A short-term surge in the tumor-infiltrating lymphocytes, particularly CD8+ T cells, as well as systemic pro-inflammatory cytokines, was seen in 11 out of 12 patients treated with ONCOS-12. Two patients exhibited both tumor-specific CD8+ T cell infiltration and systemic stimulation of tumor-specific CD8+ T cells.

IL-12 promotes the production of tumor-specific CTLs and the activation of NK cells and T cells, hence initiating anti-tumor immune responses [108]. IL-12 therapy is often regarded as a highly effective treatment for tumor-induced immune suppression, but the anti-tumor efficacy was unsatisfactory due to inadequate IL-12 delivery to the TME, lymphocytes depletion in the TME, or systemic IL-12 accumulation that could lead to potentially toxic inflammatory responses [109,110]. Thus, OAV administration seems essential for raising the level of IL-12 that targets tumors while minimizing toxicity. For example, Wang et al. (2017) formulated a tumor-targeted OAV (Ad-TD-nsIL12) to deliver nonsecreting IL-12 to tumor cells, which significantly improved the survival of Syrian hamster pancreatic ductal adenocarcinoma while reducing off-target toxicity and minimizing the spreading of IL-2 to the local TME [61]. A recent study has shown that an adenoviral vector harboring the RheoSwitch Therapeutic System (RTS) comprising the murine IL-12 gene (Ad-RTS-mIL-12) and regulated by the oral activator ligand veledimex (VDX) may extend survival in an orthotopic glioma model [111]. In a multicenter phase 1 dose-escalation trial (NCT02026271), it was shown that intratumoral injection of Ad-RTS-mIL12 and oral VDX induces local IL-12 expression dose-dependently and are associated with an increase in TIL [43].

IL-2, which has been used for the treatment of metastatic melanoma and renal cell carcinoma, is the prevalent component required for T cell proliferation and differentiation [112]. TNF-α is a multifunctional cytotoxic agent that can trigger apoptosis and necrosis in tumor cells as well as increase the production of other cytokines and activate immune cells [113]. Havunen et al. (2017) constructed Ad5/3-E2F-d24-based OAVs (TILT-123 or Ad5/3-E2F-d24-TNFa-IRES-hIL2) expressing human IL-2, and TNF-α was utilized to enhance adoptive TIL transfer. Combining TILT-123 and TILs cured all malignancies in animal models with immune competence [47]. On day 90, following the administration of TILT-123 into mouse melanoma models concurrently receiving anti-PD-1 treatment, all tumors were completely eradicated, and all animals remained alive [114].

### 6.2. OAVs Armed with Costimulatory Molecules

The absence of costimulatory molecules on the surface of cancer cells suppresses immunity in TME. Thus, targeting costimulatory pathways, such as CD40 and 4-1BB, appears to be an intriguing strategy for enhancing anti-tumor immunity [115]. Costimulatory receptor CD40 is expressed on APCs, including B cells, macrophages, and DCs. It has been demonstrated that the association between CD40 and its ligand CD40L enhances the release of cytokines, accelerates MHC class II-dependent antigen presentation, increases T cell priming, and induces cancer cell death [116]. Zafar et al. (2018) reported that an adenoviral vector expressing CD40L (Ad3-hTERT-CMV-hCD40L) enhanced the activation of DCs, which in turn led to the induction of Th1 immune responses. The OV inhibited the expression of CD40L on cancer cells, hence minimizing systemic exposure and suppressing the systemic immune response [117]. LOAd703 is an adenovirus Ad5/35 that is associated with costimulatory factors CD40L and 4-1BBL to activate the CD40 and 4-1BB pathways that elicit efficient anti-tumor immune responses. In a mouse xenograft model of multiple myeloma, LOAd703 was recently shown to enhance the activation of cytotoxic T cells and suppress tumor development [118]. LOAd703 enhanced cytotoxic T cell activation in a co-culture of immune and multiple myeloma cells, as evidenced by increased CD69, CD107a, and IFN-γ expression. In addition, they reported that LOAd703 enhanced the immunogenic profile by enhancing the production of costimulatory molecules CD80, CD86, CD70, MHC molecules, death receptor Fas, and adhesion molecule ICAM-1 [118]. LOAd703 is currently being investigated in clinical trials phase I/II. The studies involved patients with pancreatic cancer (NCT02705196) and patients with a variety of cancers, including biliary, colorectal, and pancreatic, to evaluate the toxicity and tolerability, as well as monitor tumor response, immune response, viral shedding, and survival rate (NCT0322598, NCT03225989). 

Adenovirus has also been combined with OX40 (CD134), a costimulatory molecule, to enhance the efficacy of virotherapy. Antigen interaction with OX40 is required for efficient T cell activation and T cell receptor-mediated antigen-specific signal transduction [119]. The interaction between OX40 and its corresponding ligand OX40L could enhance T cell survival, induce the secretion of cytokines, and promote T cell migration [120]. A pre-clinical study by Jiang et al. (2019) with Delta-24-RGDOX, a replication-competent OAV armed with OX40L, observed stimulation and replication of tumor-specific lymphocytes in mice models of syngeneic glioma [121,122]. Local therapy with Delta-24-RGDOX induced the proliferation and migration of tumor-specific T cells, substantially enhancing the immune system and providing an abscopal anti-melanoma response [122].

### 6.3. OAVs Armed with Immune Checkpoint Inhibitors

Despite the remarkable effectiveness of immune checkpoint inhibitors (ICIs) for some tumor types, poor responses in cold tumors and systemic autoimmune effects have prompted the use of OAVs to administer ICIs. Immune checkpoints are the key regulators of the immune system and are important in maintaining self-tolerance reducing tissue damage and inhibiting an autoimmune reaction by regulating the duration and strength of the immune response [123,124]. The production of immunological checkpoint molecules suppresses the host’s immune cell function as the host cannot mount a successful anti-tumor immune response. These molecules suppress T cell activity and are expressed on cells in the TME [125]. To avoid being eliminated by the immune system’s immunological checkpoint pathways, tumor cells disguise themselves as regular components of the human body [126]. Thus, ICIs could inactivate the immune system’s brake and trigger anti-tumor immune responses [127]. By inhibiting the immunological checkpoint to enhance T cell-mediated immune response against tumor cells, ICIs have considerably improved clinical outcomes in several malignancies. Anti-cytotoxic T lymphocyte-associated antigen 4 (CTLA-4), anti-programmed cell death (PD-1), and its ligand, PD-L1, are the most notable checkpoint inhibitors used in ICI therapy.

Zhang et al. (2019) designed an OAV (Ad5-PC) that released a soluble fusion protein with the extracellular domains of PD-1 and CD137L at each terminus. Ad5-PC demonstrated significantly enhanced anti-tumor activity in both ascitic and subcutaneous hepatocellular carcinoma (HCC) tumor models, with survival rates improved by 70% and 60%, respectively, and a sustained high level of T lymphocyte activation and IFN-γ the TME [128]. In addition, anti-CTLA-4 monoclonal armed with Ad5/3-24-CTLA and SKL002119 enhanced anti-tumor activity in lung and prostate mouse xenograft tumor models without affecting antiviral immunity or viral titers [129,130]. 

PD-1/PD-L1 is an inhibitory checkpoint that modulates peripheral T cell activity. Tanoue et al. (2017) demonstrated that OAV conjugated with helper-Ad expressing a PD-L1 blocking mini-antibody augments the anti-tumor effects of adoptively transplanted chimeric antigen receptor T (CAR-T) cells [131]. Local expression of the PD-L1-blocking mini body was preferable to systemic injection of PD-L1 IgG, confirming the unique advantage of viral delivery. A comparable method using helper-dependent adenoviruses for production of the PD-L1 blocking mini-antibody and IL-12-p70 for immune activation increased the activity and durability of CAR T cells in mice models of head and neck squamous cell carcinoma (HNSCC) [132]. In vivo results from Garofalo et al. (2021) showed that the combination of AdV-D24-ICOSL-CD40L and anti-PD-1 reduces tumor volume and increases survival by one hundred percent, indicating enhanced effectiveness and survival due to the complementary anti-tumor effects of these drugs in melanoma therapy [133]. Overall, the new oncolytic vector AdV-D24-ICOSL-CD40L, either individually or in combination with anti-cancer treatments such as checkpoint inhibitors, could offer promising therapeutic strategies for the treatment of melanoma.

### 6.4. OAVs Armed with Bispecific T cell Engager (BiTE) Molecule

Bispecific T cell engager (BiTE) molecules are the most common immunotherapeutic agents that could induce the direct cytolytic effect of T cells to lyse target cells independently of MHC expression. The BiTE molecule comprises two arms, one of which binds to CD3-epsilon on the T cell receptor and the other to a particular target antigen. Both arms, adhering to their respective target antigens, stimulate T cell activation, which induces apoptosis in target cells [134]. Freedman et al. (2017) armed OAV with a BiTE molecule that adheres to the epithelial cell adhesion molecule (EpCAM) on target cancer cells and bridges them to CD3 on T cells (EnAd-SA-EpCAM BiTE), enabling both CD4+ and CD8+ T cells to cluster and become activated. Interestingly, EnAd-SA-EpCAM BiTE demonstrated tumor cell death in the immunosuppressive TME of liquid cancer biopsies [133]. NG-641 is one of the recent developments of oncolytic adenovirus group B that incorporates BiTE molecules. NG-641 is a modified variant of the adenovirus enadenotucirev, an Ad11p/Ad3, that still exhibits all of enadenotucirev functional characteristics while also harboring transgenes that aim to disrupt the stromal barrier and impair the immune system in the tumor microenvironment [69]. NG-641 encoding fibroblast activation protein-targeting bispecific T cell activators (FAP-TAc) and immunological enhancers (IFN-α, CXCL9, and CXCL10) further enhance T cell activation by arming BiTE molecules. The production of FAP-Tac by virus-infected tumor cells ought to result in T cell-mediated destruction of CAFs, thereby altering TME to promote efficient anti-tumor immunity. Immune enhancer molecules, including IFN-α, CXCL9, and CXCL10, were developed to be encoded by NG-641 to enhance the activity of tumors with inadequate immune cell infiltration [69]. In 2020, NG-641 safety and tolerability in patients with metastatic or advanced epithelial tumors began to be studied. This phase I a/b study examined increasing intravenous or intramuscular doses in several epithelial cancer types before comparing the safety and effectiveness of single therapy or in combinations with checkpoint inhibitors and/or chemotherapy agents and is ongoing (NCT04053283). Together, the armed OAVs mediated anti-tumor strategies are highlighted in Figure 2.

## 7. Methodology

The literature selection was conducted by performing searches in PubMed (2012–2022) and Google Scholar (2012–2022) using a series of keywords, terms, and subject headings consisting of “Adenovirus, Cancer, and/or Gene Therapy”. Selected articles were evaluated by the authors based on certain criteria such as completeness and robustness of the relevant information that was available from the articles, including articles reporting case reports, experimental studies, and clinical trials. Relevant data were collected from the selected articles followed by the organization of the data into the manuscript.

## 8. Conclusions

Adenovirus has many benefits over other vectors, including effective gene transfer, large transgene capacity, and low pathogenicity. The strategic design of the OAVs aims to address key challenges associated with tumor biology, including chemo-drug resistant tumors and the protective barrier function of the TME. The OAVs that induce the activation of immune system and the recruitment of anti-tumor T cells are likely to provide tangible benefits in combating chemo-drug resistant tumors. The destruction of the TME by the OAVs and small molecule drugs offers added benefits to enhance the treatment efficacy of the OAVs. However, key challenges remain to be solved, including whether the OAVs could effectively eliminate the tumor stem cells, which are known to undergo cellular differentiation in response to anti-cancer treatment, leading to the generation of chemo-drug resistant tumor cells and metastasis of the tumor cells. Future research investigations are required to address these issues, including the design of OAVs to specifically target tumor stem cells. 

## Figures and Tables

**Figure 1 pathogens-11-01146-f001:**
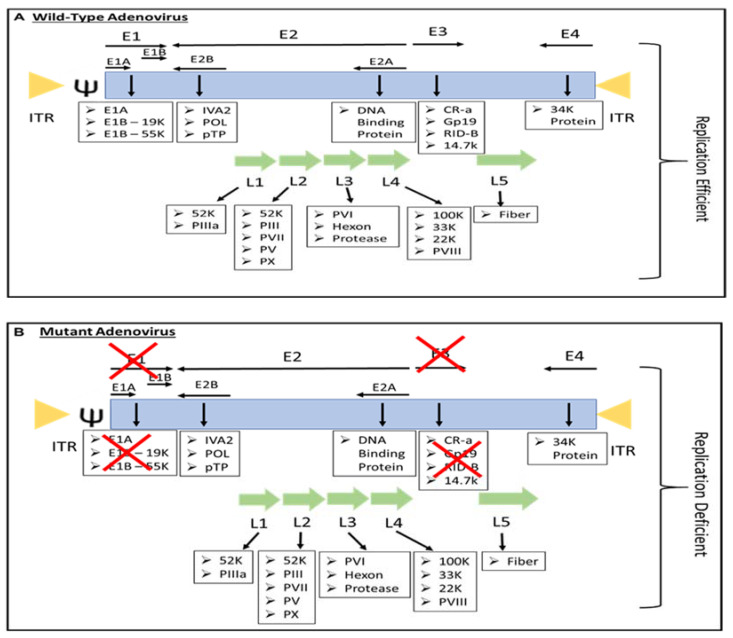
The genome of (**A**) replication-efficient adenovirus and (**B**) replication-deficient adenovirus. ITR—inverted terminal repeat; Ψ—viral packaging element; K—glycoprotein; P—pre-protein, X—Deleted.

**Figure 2 pathogens-11-01146-f002:**
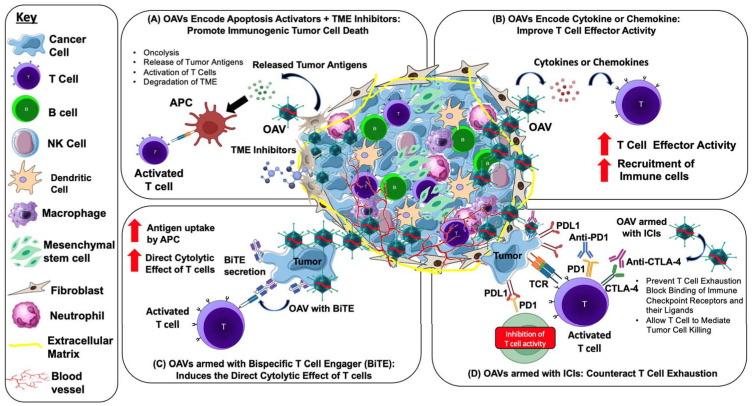
Strategies and mechanism of action for using armed OAVs to target the TME and enhance anti-tumor activity. OAVs selectively infect and replicate in the tumor. (**A**) Oncolysis caused by OAVs via apoptosis or immunogenic cell death (ICD) releases multiple molecules such as pathogen-associated molecular patterns (PAMPs), danger-associated molecular pattern signals (DAMPs), and tumor-associated antigens (TAAs), which are taken up by antigen-presenting cells (APCs) for T cell activation and ICD. The infection of OAVs is facilitated by the degradation of TME by small molecules’ TME inhibitors. (**B**) OAVs can be engineered to encode transgenes such as cytokines and chemokines, enabling targeted delivery to the TME and further activation of the anti-tumor immune response. OAVs encoding chemokines promote the secretion of chemokines into the TME that induces T cell trafficking to tumors, while OAVs encoding cytokines enhance effector function and maintain T cell survival and expansion. (**C**) OAVs encoding bispecific T cell engager (BiTE) induce lysis of tumor cells by redirecting T cells to tumor surface antigens, bidirectionally connecting T cells and tumor cells. (**D**) Mechanism of OAV combined with immune checkpoint inhibitor (ICI) antibodies. OAVs lyse tumor cells and attract CD8+ T cells into the TME, thus transforming a “cold tumor” into a “hot tumor”. Treatment with anti-PD-1/PD-L1 or anti-CTLA-4 antibodies prevents T cell exhaustion by blocking the binding of immune checkpoint receptors and their ligands such as PD-1/PD-L1 and allowing the T cell to mediate tumor cell killing.

## Data Availability

Data are available upon request by contacting the corresponding author.

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
