# Peer review of "Engineered Oncolytic Adenoviruses: An Emerging Approach for Cancer Therapy"

_pathogens, 2022, doi:10.3390/pathogens11101146_

Round 1
Reviewer 1 Report
The authors provide a review on the use of adenoviruses for the treatment of cancer, with special focus on replication-competent (oncolytic) adenoviruses for cancer immunotherapy. This is an interesting topic, in which clear description of the rationale and mechanism of action of each agent is desirable.
In general, the manuscript requires a thorough revision in order to improve the organization of concepts and the accuracy of data presented.
Specific comments:
1. The title is misleading. It should mention that the review focuses on cancer therapy.
2. A clear explanation of the different adenoviral vector versions is needed: replication-competent (oncolytic) versus replication-defective. Among replication-defective, E1/E3-deleted and High-Capacity (“gutless”) should be differentiated. Examples for the use of these adenoviral categories against cancer should be mentioned. In the current manuscript, some replication-defective adenoviral vectors such as Ad-RTS-hIL-12 are included in the table of oncolytic adenoviruses.
3. The organization of sections does not follow a clear rationale. Why the modification of tumor microenvironment is separated from oncolytic adenoviruses expressing cytokines and chemokines? What is the conceptual difference between sections 7.1 and 7.2 ?
4. Line 11 (Abstract). Chemotherapy is usually not delivered “invasively”. Maybe the authors mean “aggressively”.
5. Line 12 (Abstract). Naturally occurring adenoviruses are not used as therapeutic agents. They need to be modified in order to acquire cancer specificity.
6. Line 56. “7 hours after transfection”. Maybe the authors mean “infection”.
7. Line 82. What is “the capsid protein”? Maybe the authors mean the penton, hexon and fiber as major capsid proteins.
8. Apart from human adenoviruses, agents isolated from other species can be used as therapeutic vectors against cancer, although the current experience is very limited.
9. Line 126: “Replicative adenovirus is more effective for gene therapy [27]”. This is just an opinion, not supported by a specific study.
10. Table 1. The scope is too wide. What is the criteria for including some examples and not others?
11. Table 1. In the “type of Adenovirus” column, “Replication of defective Adenovirus Ad5” should be changed by “Replication-defective…”
12. I recommend the more generally accepted abbreviation OAV for oncolytic adenoviruses, rather than oAds.
13. Line 182. “Adenovirus expresses immune modulators and cytokines” should be changed by “Adenovirus expressING immune modulators and cytokines”.
14. Line 212. The authors state that AAV show “surprisingly low protein expression levels”. I am afraid this cannot be generalized, and depends on many different factors.
15. When comparing E1/E3-deleted adenoviral vectors with AAVs, it is not surprising that the expression is more stable in AAVs. Because E1/E3-deleted adenoviral vectors retain viral genes. The most relevant comparison should be done between AAVs and high-capacity adenoviral vectors.
16. Line 284. “more susceptible to infection” should be changed by “more susceptible to destruction”
17. Line 315. “Therefore, by substituting the virus with lymphodepletion, effectiveness was not only preserved but enhanced” should be changed by “Therefore, by substituting lymphodepletion with the virus, effectiveness was not only preserved but enhanced”
18. Line 467. “Thus, ICIs could activate the immune system’s brake” should be changed by “Thus, ICIs could INactivate the immune system’s brake”
19. The NG-641 agent is not properly described. Its function depends on different modifications apart from the transgenes. Reference #130 does not apply to this agent.
Reviewer 2 Report
The authors summarized the oncolytic adenovirus and adeno-associated virus as gene delivery vectors. The aspects are well classified and described deep enough. Therefore, I recommend this manuscript for the publish as it is.
Reviewer 3 Report
A very interesting and complete rewiew about the possibility of using adenovireses for gene therapy of oncological deseases is presented. The structure and immunogenicity of adenoviruses is described in great detail. Also described their genetic modifications. Successes and prospects of their use in the treatment of a number of tumors completely described. The article will be useful for virologists, immunologists and clinicals. The article may be published withoutsignificant changes.
Reviewer 4 Report
With interest I read the manuscript Adenovirus: An Emerging Approach for Gene Therapy by Ee Wern Tan and colleagues. The manuscript is well written and the topic of the review is timely. There are a number of major aspects that need attention.
1) The section 6 ‘Adenoviruses versus AAV’ should be omitted. AAV and Adenovirus have really very little in common (apart from, maybe, their names). No one would be in doubt which of the two to choose for a particular gene therapy application. If one would compare adenoviruses with their clinical rivals, Herpes Simplex virus, Vaccinia virus or even Myxoma virus would be far more obvious choices for use in oncolytic virus strategies. AAV is a very odd choice for coverage in this review.
2) This reviewer would have liked to see a more focused treatment of the topic, in which the review is used to bring the reader to a conclusion. As it stands now, the factual information is compiled and organized, but not used as a flow of arguments that leads to a point. This is evident from the Conclusions section, that only makes a number of very general and broad statements. Could you rewrite parts of the text to guide the reader to a conclusion? Now the statements of the Conclusion section merely reiterate the info of the Introduction section.
3) The low pathogenicity of adenoviruses is frequently stressed (e.g. line 536). It would be prudent to at least mention its involvement as ocular pathogen, or the implication of HAdV-F in severe diarrhea in children.
4) Adenovirus has not yet emerged as an alternative treatment against cancer (contra line 537) despite the high hopes and promising results that the field has. To reach this stage more treatments need to be formally approved and registered. Please tune down this phrase.
5) The treatment of the serotypes should be updated. At the moment approximately 114 serotypes and types have been approved by the human adenovirus working group. Also it would be useful to explain the terms type (largely based on genomics data) and serotype (largely based on serology) that are now both used in literature.
6) The seroprevalence listed for Ad3 (line 80) seems erroneous and this reviewer could not find such data in ref 10. Please check.
7) The statement of line 159-160 is potentially misleading, and the data are not in ref 47. Ref 47 states explicitly: 'However, the true impact of this new immunotherapy on overall survival will only be determined in a statistically controlled and randomized trial with larger numbers of patients.' So please tune down this statement.
8) Could the authors include a statement how they identified and selected the literature covered in this review?
Minor points
9) Ref 28 seems incomplete
Round 2
Reviewer 1 Report
Second revision
Information about High-Capacity Adenoviral vectors remains disorganized. Section 4 starts talking about these vectors, but they are not properly described until line 139.
I suggest elimination of section 4 and inclusion of the information related to High-Capacity Adenoviral vectors in section 5, whereas information about immune responses can be allocated in section 3.
Reviewer 4 Report
The text is much better structured and to the point, the responses to the reviewers comments are appropriate and improved the manuscript.
Some minor typo's crept in that should be corrected.
Line 11-12: Please amend ‘Unfortunately, chemotherapy is frequently delivered aggressively and are invariably accompanied by dose-dependent cytotoxic side effects.’ To read ‘Unfortunately, chemotherapy is invariably accompanied by dose-dependent cytotoxic side effects.’ (In our hospital we tend not to deliver chemotherapy aggressively …)
Line 85: Please amend ‘strains F40 and F41’ to read ‘types F40 and F41’.
Line 163: Please amend ‘Kim et.al has reported’ to read ‘Kim et al. have reported’
